# Community Assessment for a Low-Carbohydrate Nutrition Education Program in South Africa

**DOI:** 10.3390/nu15010067

**Published:** 2022-12-23

**Authors:** Georgina Pujol-Busquets, James Smith, Sergi Fàbregues, Anna Bach-Faig, Kate Larmuth

**Affiliations:** 1Research Center for Health through Physical Activity, Lifestyle and Sport, Division of Physiology, Department of Human Biology, Faculty of Health Sciences, University of Cape Town, Cape Town 7700, South Africa; 2Faculty of Health Sciences, Universitat Oberta de Catalunya (Open University of Catalonia, UOC), 08018 Barcelona, Spain; 3Department of Psychology and Education, Universitat Oberta de Catalunya (UOC), 08018 Barcelona, Spain; 4FoodLab Research Group (2017SGR 83), Faculty of Health Sciences, Universitat Oberta de Catalunya (Open University of Catalonia, UOC), 08018 Barcelona, Spain; 5Food and Nutrition Area, Barcelona Official College of Pharmacists, 08009 Barcelona, Spain

**Keywords:** nutrition education, community assessment, dietary strategy, qualitative study, low-carbohydrate, South Africa, under-resourced communities

## Abstract

Eat Better South Africa (EBSA) is an organization that provides low-carbohydrate, high-fat (LCHF) nutrition and health education programs for women from under-resourced South African communities. Community assessments are essential to explore participants’ potential facilitators and challenges of adhering to new dietary behaviours and should be implemented before any dietary interventions. This study is a qualitative community assessment to enable the EBSA program to better meet potential participants’ needs and explore their willingness to enrol in the EBSA program. Sixty women from two communities in the Western Cape were interviewed through six focus group discussions. A thematic analysis was conducted using NVivo 12 software, and four themes were developed around the women’s (1) role within the households; (2) dietary behaviour; (3) health perceptions; and (4) willingness to participate in an LCHF program. Women mentioned that they were responsible for cooking and shopping for their households. They expressed their understanding of healthy and unhealthy behaviours and their dietary patterns. Some women showed concerns about LCHF diets, but others wanted to learn more due to their knowledge of other people’s positive experiences with the diet. There was a general desire to become healthy. However, the women anticipated dietary behaviour change to be challenging. Those challenges mostly revolved around their socioeconomic environments. The findings are intended to inform EBSA (or other nutrition interventions) on what to consider when implementing their interventions in these communities.

## 1. Introduction

The non-communicable disease (NCD) epidemic is one of the 21st century’s major challenges regarding global development [1,2]. While NCD rates have rapidly increased in South Africa, communicable diseases (CDs) persist, primarily because of the prevalence of the human immunodeficiency virus (HIV) and tuberculosis (TB) [3,4,5,6]. While individual behaviours contribute significantly to the growing burden of NCDs [7,8], in underserved communities, contextual factors such as access to education and healthcare, food insecurity, and poor-quality food limit certain healthy lifestyle choices [9,10]. In these communities in South Africa, most of the population purchases and consumes staple foods (mainly maize, rice, and bread) due to their affordability. These foods are highly subsidized by the South African government, where no value-added tax (VAT) is paid on these foods. These staples cost less per unit of energy than animal products, fruit, and vegetables [11,12,13] and are the preferred food choices of most people in more impoverished communities [14,15].

Women fulfil many demanding nurturing, family-related, and societal roles [16,17], frequently disregarding their health to address that of the people around them [18,19]. Therefore, it is common to find women purchasing these staple foods when shopping for their families within these communities [14]. These foods typically contain high quantities of refined starch, with simple sugars and added fats or oil to enhance flavour and improve their satiety levels [20]. All of this can lead to “hidden hunger”, when people eat regularly and even gain weight but lack the necessary nutrients and vitamins, leading to reduced health and increased disease states [21].

The prevention and mitigation of NCDs in these communities may be positively supported through engagement in community-based nutritional education programs [10]. Essential elements for a successful community-based program include a good understanding of community functions, close collaboration with various community organizations, and the full participation of the people themselves [22]. Qualitative methods have increasingly been used to assess a target population’s needs and preferences and to evaluate given education programs [23,24,25]. Focus group discussions (FGDs) have been used successfully to develop, or adapt, self-management interventions for NCDs, especially in low-income populations and minority groups [25]. The informal style and interactive nature of an FGD are particularly appropriate for identifying barriers to care, exploring health beliefs, identifying education needs, and gathering information to improve intervention programs [23,24,25]. 

Eat Better South Africa (EBSA) is a non-profit organization that has been running six-week nutritional education programs for people—primarily women—from under-resourced South African communities since 2015 to prevent or treat diet-related NCDs [26]. EBSA has based its nutritional recommendations on low-carbohydrate, high-fat (LCHF) diets to treat or prevent metabolic health conditions. A major motivating factor for women to turn to LCHF diets is to lose excess body fat [26], since it increases fat metabolism and decreases hunger-promoting hormones [27]. With the onset of menopause, increases in body fat and central adiposity are associated with a decrease in hormonal activity [28,29,30]. Initially, LCHF diets became popular in South Africa to lose weight and to improve metabolic health [27,31,32]. There is a general perception that the foods included in an LCHF diet are expensive and that most of the people in South Africa who decide to follow this lifestyle are reasonably affluent, and it is not applicable to under-resourced communities [33,34,35]. However, it is people from these under-resourced communities who suffer the highest rates of illness from NCDs due to limited access to education and healthcare [36,37]. The EBSA program has helped women from two communities—Atlantis and Ocean View—to lose weight and improve their metabolic health; however, the program was lacking in certain areas, and improvements were suggested [26]. Briefly, members found that they needed greater ongoing support from EBSA, that LCHF food substitute suggestions were not applicable, and that some of the motivational tools the program employed (e.g., “the biggest looser”—like a weight loss competition) were counterproductive [26]. The literature further highlights the need to assess a community’s preferences and needs prior to the implementation of any community intervention [23,24,26]. 

To date, only one study has investigated participants’ perceptions of and experiences with an LCHF dietary intervention in South African communities where socioeconomic challenges such as poverty, unemployment, and crime are prevalent [26]. These same communities suffer from a high prevalence of obesity and a disproportionate burden of NCDs. Indeed, qualitative assessments can be a tool for involving the community in solving problems and developing goals before taking any action [23,24]. The work presented here is a qualitative community assessment of two under-resourced South African communities with women naive to the EBSA program. This study explores women’s needs and preferences and their willingness to participate in LCHF programs such as EBSA to improve their health. The hope is that these findings will help interventions to better meet participants’ needs and identify ways to improve the recruitment for, acceptance of, compliance with, and sustainability of future LCHF education programs in these communities.

## 2. Materials and Methods

### 2.1. Study Design and Ethical Approval

This was a community assessment that used 6 FGDs with 60 women who had never taken part in an EBSA program. Ethical approval (HREC REF 391/2018) was granted by the Faculty of Health Science Human Research Ethics Committee of the University of Cape Town, and written informed consent was obtained from all participants. The study was conducted in accordance with the ethical principles of the Declaration of Helsinki.

### 2.2. Study Settings

The study took place in economically disadvantaged communities in the Western Cape Province of South Africa, situated approximately 50 km from the province’s major city of Cape Town. These communities emerged and originated from the Group Areas Act under the apartheid government [38]. The cohort of women were recruited from Ocean View and Cloetesville in 2018. Ocean View has a population of nearly 14,000 people, with approximately 3083 households. Just over half (51%) of the community are women, and 91% are of mixed-raced ancestry. Afrikaans is the most widely spoken language and is the first language for 57% of the population, followed by English at 39% [39,40]. 

Cloetesville was founded in 1964 and is a community located adjacent to the township called Kayamandi in Stellenbosch. It is surrounded by the Cape Winelands’ vineyards and the mountainous nature reserves of Jonkershoek and Simonsberg. The community has 16,000 inhabitants and is served by one public health clinic. Approximately 52% of the community are women, and 88% are of mixed-race ancestry. Afrikaans is the most widely used language and is the first language for 94% of the population, followed by English at 3%. Cloetesville was developed as a farmworkers’ settlement area, close to the farms. In essence, the development served to provide labour for farms and the town [39,40].

### 2.3. Participants

Participants were recruited at a community meeting with the help of a non-profit organization working with members of the community. There were two community meetings, one for each community. Purposive criterion sampling [41] was used to select participants fulfilling the following eligibility criteria: women; 18 years or older; capable of providing informed consent; able to understand and speak English or Afrikaans; and able to attend the focus group meeting. Women who were interested in taking part in the study were given a participant information sheet, and those who met the eligibility criteria were recruited as part of the study sample.

### 2.4. Procedures

Six FGDs were conducted between September and October of 2018 and included a minimum of 6 and a maximum of 11 women, with 1 moderator. All 6 FGDs were held on different dates, and there was only 1 group per date: groups did not repeat sessions. The discussions lasted, on average, 1 h and 50 min. In accordance with recommendations in the literature on the sample size needed in a qualitative study to achieve saturation, our relatively homogeneous sample was large enough to identify 90% of the common themes [42,43]. FGDs were held in a venue within the community. Upon arrival, each participant reviewed and signed an informed consent form and completed a sociodemographic questionnaire in the presence of a member of the research team who was fluent in the participant’s preferred language (English or Afrikaans). 

A bilingual independent (Afrikaans and English) experienced FGD facilitator and researcher in qualitative studies was hired to moderate the FGDs. Participants were asked to express themselves in the language they felt most comfortable in, predominantly Afrikaans. The moderator encouraged conversation among participants using ten open-ended questions developed by the researchers for this study. These questions were not necessarily asked verbatim (Appendix A). The themes of the questions were related to women’s current dietary patterns, facilitators and challenges of dietary behaviour change, and knowledge of LCHF diets and EBSA. Prompts and probes were used, when necessary, to elicit further information. During the focus group, a researcher who assisted the moderator took notes, which were used to create context and give clear and consistent discussion details. Refreshments were provided at each focus group, and participants were reimbursed as compensation for their time.

### 2.5. Data Analysis

FGDs were audio-recorded in the participants’ language (Afrikaans) and were transcribed and translated into English by Cyber Transcription (South Africa). Using NVivo 12, Braun and Clarke’s approach to thematic analysis [44,45,46] was implemented in six phases [44]. In Phase 1, Georgina Pujol-Busquets (GPB) read the transcripts several times to familiarize herself with the data and identify initial patterns. Based on these readings, in Phase 2, GPB developed a first draft of the codebook, which was subsequently revised by Anna Bach-Faig (ABF) and Sergi Fàbregues (SF). Phase 3 involved GPB searching for meaning in the data by dividing each code into several subcodes. In Phase 4, subcodes were reviewed by ABF and SF and subsequently displayed by GPB as thematic maps. Phase 5 consisted of naming the themes and defining the main concepts of each theme, with a total of four. Phase 6, mainly carried out by GPB, Kate Larmuth (KL), and James Smith (JS), consisted of producing a report that was later shared with the EBSA members and that included recommendations on how to optimize the program for future interventions. Although only GPB was involved in the coding, all members of the research team reviewed the coded data and agreed on the final categorization and themes to minimize personal bias.

## 3. Results

A total of 60 women completed the study by taking part in 6 FGDs, with 6 to 11 women per focus group. All participants reported that they had mixed-race ancestry. The mean age was 58.1 ± 13 years, and the mean body mass index (BMI) was 31.1 ± 6.8 kg/m^2^. Table 1 shows the characteristics of the 60 study participants. 

Four themes were developed from the analysis of the participants’ FGDs (Table 2): theme 1: role within the households; theme 2: dietary behaviour; theme 3: health perceptions; and theme 4: willingness to participate in a LCHF program.

### 3.1. Theme 1: Role within the household

Women described their roles within their households and in the community as that of caregivers. Women mentioned that it is their duty to take care of both children and husbands, which often entailed different roles.

“I am a mother and a wife and the men do not think the way that I think. They do not worry. I always think about what my children and husband are going to eat.” (Ocean View)

Tradition was something that women mentioned quite often when describing cooking roles and their roles in their households. They reported that it was expected of them, as caregivers, to cook for the family, because that is how they were raised and what they saw when growing up. Although the women themselves said they were doing most of the cooking in their homes, sometimes they shared their cooking duties with other women in the household. Many women mentioned that it is expected of them to cook large meals to feed everyone in the family.

“To prepare a complete cooked meal for your children or family is tradition. We grew up with that and that is how your mother taught you.” (Cloetesville)

Participants reported that they added starches, such as potatoes, rice, and bread, to their family meals to increase the volume, particularly in food-insecure households and when there were men living in the home.

“Sometimes when the food is not enough, then you have to stretch the food. (…) We make use of potatoes and rice; it is relatively cheap, and we have big families.” (Ocean View)

Even though cooking was identified as a woman’s responsibility, the women mentioned that, most of the time, men dictate the type of food they buy. In addition, monetary issues were mentioned by some women, as some depend on their husbands’ budgets to buy and prepare their meals.

“I have three men in the house and my children are big and they eat a lot. My husband lost his leg because of sugar. He was the breadwinner, and the problem is that there is no money.” (Ocean View)

When shopping, women considered money and distance to their homes as priorities when choosing the stores they shop from. They chose shops that were either vastly cheaper (but perhaps a bit further away) or just those that were most convenient (on the way to/from work or close to the house). They also chose where to buy their groceries depending on the amount of money they could spend. Moreover, they mentioned that they buy food that is faster and easier to cook, because the time to prepare food is constrained by work schedules.

“The food that I prepare at home must be fast dishes, because of a person’s working circumstances. Moreover, quick things because it is easy.” (Ocean View)

### 3.2. Theme 2: Dietary Behaviour

At most of the discussions, participants said that they ate three meals per day. In the morning, they ate cereal, oats, or mealie meal with sugar and milk, and tea or coffee. A common lunch reported by participants was a sandwich with cheese, Marmite, peanut butter, or jam. Some participants mentioned that they ate whatever was in their cupboards, which meant that they sometimes only had bread at all three mealtimes. Dinner was the main meal and usually consisted of rice, potatoes, and sometimes chicken. The participants said that as they usually must cook for many people, they avoid cooking meat, since it can be very expensive. A few participants said that they enjoyed koeksisters (sweet doughnuts) on weekends; others said that they baked cake every weekend, and one participant said that she enjoyed eating biscuits. They also pointed out that they had braais (barbecues), which they all felt was a very healthy way of preparing food.

“Mine [diet] consists of bread. In the morning, I will have porridge, such as Jungle Oats and Weet-Bix.” (Ocean View)

Some talked about their cravings and downfalls, usually regarding sweets and chips. Women mentioned that they get triggered by stress and anxiety, and that was what usually led them to eat those foods. In addition, not being in control of their eating habits was something the women repeatedly mentioned when being asked about their dietary choices and preferences around food.

“Late in the evening when everyone is sleeping, then you go on a hunting trip to find something to eat, chips or something else that you know is going to make your joints pain or make you sick. I feel that I have a psychological problem.” (Cloetesville)

When talking about “must have” food items, the women mentioned several things, but mostly starchy foods. Participants identified their food choices as crisps, potatoes, bread (toast with butter and jam, cheese, Marmite or butter, margarine, and peanut butter), rice, and some vegetables, such as sweet potatoes, cabbage, and pumpkin, with their dinners. In terms of drink preferences, the women mentioned a variety of drinks, especially coffee. Some felt that water was not something that they enjoyed and said that they were satisfied with consuming water in the form of tea or coffee. Some participants expressed that they drink alcohol (beer, red wine, and whiskey) on weekends, and others said that they drink sugar-sweetened drinks.

“I am obsessed with coffee. (…) I drink up to nine cups of coffee a day. I can stay without food for the whole day as long as I have my coffee.” (Ocean View)

### 3.3. Theme 3: Health Perceptions

When women were asked about what they understood as being healthy, some mentioned aspects related to diet and the types of foods considered healthy. Others mentioned that healthy living means making healthy choices, which depend on everyone. They also mentioned eating with restraint or portion control and focusing on healthy food preparation.

“I think that a healthy lifestyle includes that you have to eat healthy every day; you need to exercise and eat a green salad.” (Ocean View)

When describing what they thought was healthy, one participant mentioned that she uses the brand Futurelife (Durban, South Africa), while others said that they considered low-fat products healthy. Futurelife is a South African food company that sells cereals, drinks, and snacks high in carbohydrates and low in fats as healthy options for the public. Most of the participants understood a healthy lifestyle to mean three meals per day; drinking a lot of water, especially first thing in the morning; cutting down on oils and fat; eating fresh fruit and vegetables; incorporating exercise into their daily routine; and getting enough sleep. Some participants felt that inner joy was important for one’s health.

“It [healthy living] comes down to correct eating habits, enough exercise, to drink enough water and to stay calm.” (Cloetesville)

“For me it is important that you get enough sleep, your stress levels need to decrease, and you have to develop balanced eating habits.” (Ocean View)

When the women were asked what an unhealthy diet meant to them, they mentioned foods high in dietary fats, starchy foods, and takeaways. Participants associated chronic illnesses such as diabetes, heart disease, arthritis, gout, and high blood pressure with unhealthy eating, such as cooking with a lot of oil and a lack of exercise.

“Can I say everything that is white? [laughter] Rice, sugar, potatoes, even mealie meal and sweet potato.” (Cloetesville)

Moreover, some mentioned that eating too much of a certain type of food is unhealthy and that alcohol can also be harmful.

“Too much of one thing is unhealthy. (…) I also think that alcohol plays a role. It is not wrong to drink, but a certain percentage of alcohol can be unhealthy.” (Ocean View)

In terms of ways to improve their health, they said that incorporating fresh vegetables and fruit into their meals was good for their health, that steaming and grilling their food was healthier than cooking with a lot of oil and fat, and that cutting down or reducing sugar in sauces was healthy.

“At the moment I try to reduce the carbs and sugars, I really try hard. It is tough, but I try because you always go back to your bad habits. I try every day.” (Ocean View)

Some mentioned that physical activity is important for them to have a healthy lifestyle and that they try to maintain an active lifestyle. They understood that by exercising, they could eat whatever they wanted.

“The reason that I joined the gym is because I want to eat what I want to eat. If I want to have biryani today, then I will have biryani, because tomorrow I will burn it out in the gym.” (Cloetesville)

### 3.4. Theme 4: Willingness to Participate in an LCHF Program

Most of the participants felt that their current lifestyle and eating habits were not healthy, and they were all keen to change to a healthier lifestyle. They acknowledged that this meant changing their current eating habits and incorporating more exercise into their routine.

“Just one word, healthy lifestyle, healthy eating. That is my mindset change for myself to have a lifestyle change. I want to have a healthy life.” (Ocean View)

“I think everyone sitting here this morning does not really lead a healthy lifestyle. Our eating habits are not healthy, but we try.” (Cloetesville)

As mentioned previously, the women indicated a desire to become healthier and a willingness to be enrolled in a program where they could receive education about nutrition and healthy eating habits. Two or three participants added that they were already following a low-carbohydrate eating plan that was much like Banting, as advised by their doctors. The women stated that they wanted to enrol in the program for themselves, but also to be a good example for their families, especially for their children.

“I also think that it will be good for myself, so that I can instil it in my children and especially my husband, who is a diabetic.” (Cloetesville)

Participants from Cloetesville mentioned that, most likely, the community would not be interested in a Banting/low-carbohydrate nutritional program, although they would consider it. Some women in Ocean View knew about the EBSA program because they knew of other women in the community who had previously taken part in the program. They said that the experiences of these others made them want to enrol in the EBSA program.

“For myself I need to make a dramatic change. Last year they treated me for organ failure, I almost died. (…) Therefore, for me this is important, this time I need it. It worked for my sister. She is still on Banting and she is very healthy.” (Ocean View)

The women agreed that their communities needed nutrition education and community-based interventions. Only a few participants had heard about the Banting/low-carbohydrate eating plan but did not know much about it and thought that it might be expensive to follow. Some voiced concerns about the recommended foods on LCHF diets, such as the amounts of dietary fats and the contradiction with what doctors had told them.

“I also say that the Banting diet is a little bit expensive, and I am not fond of it. According to what I have read, it is a lot of greasy food, because I can now eat bacon and eggs.” (Cloetesville)

“For me it was also shocking. The diet with the amounts of fat that we always think we need to cut out of a diet (…) We always believe what doctors tell us.” (Ocean View)

There was a general desire in the whole group to become healthy, and the women reflected on their life’s struggles and challenges to do so. Many could identify stress, poverty, and unemployment as reasons for their unhealthy food choices. They said that gang violence and safety were things they had to deal with daily. The women also mentioned that some of the challenges that they are facing would make it difficult to be able to change their habits, not only for them but also for their families. Mental health was also something participants were concerned about, and some mentioned that they felt they were not ready for such a significant dietary behaviour change.

“Mentally I am not ready to participate in any program. In the past few months, I had a setback where I lost heavily. (…) I was just in the house and I took it out on food, I just wanted to eat everything that I craved. (…) I am still dealing with my losses, so I am not mentally right to participate.” (Ocean View)

As stated before, the women mentioned that if they were about to change their food habits, their families’ habits would have to change as well, so some said this could create a huge burden on their budgets. Therefore, according to the women, tradition and culture were seen as potential challenges as well.

“I am not willing to make two dishes; hell, no, it is too expensive. Therefore, if something can work it out, how to keep everyone in the family happy.” (Cloetesville)

## 4. Discussion

This study investigated the needs and preferences of women who had never taken part in an EBSA or any other LCHF program. These participants were from two under-resourced South African communities, where residents have limited access to education, employment, and healthcare. Alcohol and drug abuse and violent crime are also prevalent within these communities [47]. To our knowledge, this is the first study to document women’s willingness to participate in an LCHF nutrition and education program in a challenging socioeconomic environment in South Africa. Given that LCHF diets are perceived as expensive and in conflict with current dietary guidelines [48,49], a major aim in this study was to see whether EBSA was feasible in these communities by exploring the factors that may impact the program, as well as the perceived barriers to and facilitators of dietary compliance.

### 4.1. Weight Loss Is a Major Motivator for Change, but Change Is Often Expensive

As is typical in other dietary intervention studies [26,34,50,51,52,53,54,55], the desire to improve their health and lose weight was crucial to this cohort when considering enrolling in the EBSA program. As evidenced in the group discussions with the women, most seemed aware of their unhealthy dietary habits. A few women even had nutritional views on health that aligned with an LCHF diet. Women generally indicated that they wanted to make the appropriate dietary changes to improve their health, although some felt that an LCHF diet would be too expensive for them, in line with the common perceptions of LCHF diets, and this was noted as something for the EBSA program to focus on. A qualitative study conducted in New Zealand with a Māori population reported that financial constraints prevented participants from adhering to an LCHF diet for an extended period [49]. On the contrary, previous EBSA participants from similar demographic communities in South Africa said they could afford the foods on the program’s meal plan [26]. This was highlighted as a possible barrier to EBSA’s implementation of the program in the current communities should the participant have less financial security. The current qualitative study did not perform any quantitative assessments of diet or financial status beyond unemployment. Therefore, it is not clear to what degree this sample represented the financial status of all potential EBSA participants and their wider communities. However, the findings on financial constraints as a barrier to the adoption of interventions are in line with those of previous studies [50,51]. Interventions should attempt to assess the financial spectrum of the communities they enter. The financial implications of the nutrition advice that EBSA provides should possibly be tiered to try to provide a spectrum of LCHF solutions (i.e., low-cost LCHF to moderate-cost LCHF advice). 

### 4.2. Family Habits and Social Support Are Major Factors in Food Choice

Generally, family support is believed to aid adherence to a range of dietary and health interventions [56,57,58,59,60]. This was emphasized in this study, as the “men in the house” had to be on board with whatever changes the women were to make. In our work with previous EBSA participants, the lack of support from family and peers was identified as a significant challenge to compliance, and it made eating LCHF food in social situations difficult [26]. This is a valid concern; similarly, diabetic patients often struggle to cope with their illness due to the absence of support from their social environment [61,62]. In the current study, adding to this existing challenge, family dietary behaviour was identified during the FGDs as a potential obstacle. Family food preferences, food selection, and preparation were anticipated by participants to be a major source of reluctance to make dietary changes. Although we did not quantitively assess the size of the participants’ families or whether they had children or not, the results here are similar to those of other studies conducted in different cultures [50,51]. To overcome these obstacles to support, it would be beneficial if interventions could, in at least one session, explain the health and financial implications of improved health and diet to family members. Nutrition interventions would also have to be considerate of the size and number of “bread winners” in each family and extend the invitation to join the program to more than one member of the family. 

### 4.3. Environmental Challenges to Health and Wellbeing in This Community

The physical environment could be seen as a barrier to the accessibility and availability of healthy foods. It relates to the distant location of supermarkets or grocery stores and the high cost of healthy foods in local shops [63,64,65]. This is an essential finding, since neighbourhood food environments can influence the target audience’s ability to implement the nutrition messages received [63,64,65]. In congruence with other studies in low-income communities [26,49,63,64], this cohort indicated that social factors such as gang violence were major challenges to following a healthy lifestyle. Unfortunately, gangsterism, drug abuse, and violent crime are endemic to the townships and communities around Cape Town [47,65]. While dietary intervention programs are unlikely to change these social issues directly, organizations should be aware of the limitations they place on participants. These include high social stress levels and a challenging environment in which to conduct physical activity [26]. The overall social environmental burden placed on people living in these communities is great [47,65]. This impacts their mental health and subsequently the women’s willingness to participate in any health intervention program. These communities have a lot to worry about, and the added stress (as it is often perceived) of taking part in, or being open to, change inhibits recruitment. It would be prudent for any nutrition intervention to try to facilitate safer spaces for healthy behaviour first and to not create a divide between those who are psychologically ready to take on re-education and those who are not. This is especially true in communities whose members strongly associate physical activity with health and stress relief [47,65]. 

### 4.4. Current Perceptions of Healthy and Traditional Eating May Influence Adoption

Some people cannot identify healthy food, nor how food relates to health outcomes [66,67,68]. However, these communities had a significant awareness of what was healthy and what was not. The women’s health awareness was largely centred around exercise, stress, and low-fat foods, the latter being a major obstacle for any LCHF intervention. Nutrition education is required to provide information about the “hidden sugars” in everyday foods that are currently contributing to the NCD epidemic. Nutrition education aims to benefit society by educating the general population about the health impact of what they eat [66,67,68]. Knowledge of the benefits of low carbohydrates and protein, healthy fats, and fibre should be within everyone’s reach. LCHF diets remain highly controversial, mainly due to increased fat consumption and reduced whole-grain consumption [49,62]. Indeed, this was seen as something that would worry women when thinking about enrolling in the EBSA program, since they thought it would be a potential challenge, and their health knowledge base implicates fats in heart disease and sees them as unhealthy. Interventions should endeavour to answer any concerns around these issues in the recruitment and enrolment period.

Cultural foods and traditions matter, and here, they were seen as potential challenges, as the role of the woman in a household (responsible for the satiation of the family in a healthy way) influences her food choices. These traditions are often connected to memories and, therefore, feelings and emotions that influence decision making [69,70,71]. Women from under-resourced South African communities come from a previously segregated context, where people were forcibly removed from their communities [38,39]. Treating culture as a static factor often furthers stigma and racial bias, which perpetuates the inequalities that attention to cultural differences is meant to redress [69,70,71]. Therefore, themes around culture and tradition are critical findings in this study and others [69,70,71,72,73] and should always be assessed prior to any intervention to better address people’s needs.

### 4.5. Intervention Recommendations

Although the findings of this article cannot be generalized, they provided insight into the potential challenges that women from the Western Cape’s under-resourced communities could face when taking part in an LCHF education program. The FGDs allowed conversations to naturally evolve, and we could not ensure that women’s responses were not influenced by others, either present in the groups or not. This is a common potential bias in qualitative research that is intended to be overcome by ensuring the expertise of the moderator. The results of this study were interpreted, and a report was written with some suggestions for the EBSA team. The recommendations were divided into four main topics, namely, recruitment, the program’s structure, follow-ups, and publicity in the community (Appendix A). 

Regarding recruitment, previously, some of the EBSA programs were offered to women who were already taking part in a physical activity group, which was free [26]. Indeed, EBSA could look for groups already created within the community to recruit participants. EBSA would benefit from collaborating with existing organizations that already work in the community. Regarding the EBSA program’s structure, safety was seen as a challenge for this study’s participants; this was true for previous EBSA participants too [26]. Therefore, it was recommended that EBSA provide participants with exercise activities to perform at home or a safe space or transportation means to do so. Moreover, women’s mental health was a major focus, as they thought their dietary habits were out of control and that stress made them eat compulsively. EBSA could create a session related to mental health, how to deal with stressful situations, and the environments they face. Hunger control related to mental health is well within their purview to help women deal with these situations. Regarding follow-up and support, the EBSA coach’s role is a necessary tool for EBSA’s sustainability within the community, as seen previously [26]. In new communities, EBSA should identify a woman among the participants who appears to have the best profile to become the next EBSA coach for the community. Considering that unemployment rates and retrenchment among women in those communities are high, it should be essential to know about women’s situations and choose someone with whom the group would feel most comfortable. 

Finally, in regard to publicity in the community, many women reported that they had little or no support from family, friends, or colleagues at the beginning of previous EBSA programs, although some experienced increased support when these people could see results [26]. Some women also tried to encourage other people to join the program or follow the diet but struggled to convince them [26]. Moreover, women from the community assessment acknowledged that one of their greatest challenges was their families’ support. By having an EBSA session where participants can invite people who are usually around them, the program and diet can be explained to these people, and by doing this, women might receive more support from their networks. 

## 5. Conclusions

Women were receptive to nutrition education interventions being run in their communities and were in need of health improvement programs. The results reveal personal and environmental factors that could make it challenging for participants to make dietary changes or to potentially adhere to LCHF dietary recommendations. Moreover, the women’s opinions around dietary behaviour are heavily influenced by their family practices. Women strongly emphasized financial problems, food insecurity, and the cost of healthy foods as barriers to health. This was an expected result, as most of the participants in this study were unemployed. This community assessment helped to uncover not only the participants’ needs but also the underlying culture and social context that could help EBSA understand how to address the community’s requirements. The numerous facilitators and barriers identified in this study are likely to be major contributing factors to future interventions’ success. Barriers such as food security, the lack of safety, and economic health are mainly in the personal, socioeconomic, and physical environment domains. Nevertheless, most participants expressed a strong desire to become healthier and learn more about nutrition and were receptive to the concept of an intervention. 

## Figures and Tables

**Table 1 nutrients-15-00067-t001:** Participants’ characteristics (*n* = 60).

**Characteristics**	** *N* ** **(%)**
EBSA program	
Ocean View	6 (10%)
Cloetesville	54 (90%)
Highest level of education completed	
Primary School	20 (33.3%)
High School	30 (50%)
Certificate	4 (6.7%)
Diploma	2 (3.3%)
Degree	4 (6.7%)
Work status	
Unemployed	47 (78.3%)
Employed	13 (21.7%)
Reported medical conditions	
High blood pressure	28 (46.7%)
High cholesterol	14 (23.3%)
Obesity	4 (6.7%)
Type 2 Diabetes Mellitus	7 (11.7%)
Heart problems	1 (1.7%)
Type 1 Diabetes Mellitus	3 (5%)
Osteoporosis	2 (3.3%)
Arthritis	4 (6.7%)
Thyroid	4 (6.7%)
Depression	1 (1.7%)
None	18 (30%)

**Table 2 nutrients-15-00067-t002:** Themes, subthemes, and definitions developed from the focus groups with participants.

Themes	Subthemes	Definition of the Themes
Theme 1: Role within the household	1.1 Women’s responsibilities 1.2 Factors influencing shopping	Women’s roles and responsibilities as caregivers. Priorities when shopping for groceries.
Theme 2: Dietary behaviour	2.1. Current diet 2.2. Feelings around food	Descriptions of current diets and feelings around foods.
Theme 3: Health perceptions	3.1. Healthy and unhealthy lifestyle 3.2. What women do to be healthy	Understanding of being healthy and unhealthy.
Theme 4: Willingness to participate in a LCHF program	3.3. Desire to be healthy 4.1. Knowledge of LCHF diets 4.2. Barriers to adhering to dietary recommendations	Willingness to participate in the EBSA program. Socioeconomic challenges to adhering to the diet.

## Data Availability

The data presented in this study are available on request from the corresponding author. The data are not publicly available due to participants’ confidentiality.

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
