# Peer review of "Community Assessment for a Low-Carbohydrate Nutrition Education Program in South Africa"

_nutrients, 2022, doi:10.3390/nu15010067_

Round 1

Reviewer 1 Report

The non-communicable disease epidemic is one of the 21st century's major challenges regarding global development. The manuscript, Community assessment for a low-carbohydrate nutrition education program in South Africa, is a qualitative community assessment of two under-resourced South African communities. The findings of this study were useful to optimize future interventions.

The aim of this study is to explore women’s needs and preferences and their willingness to participate in a low-carbohydrate high-fat program to improve their health.

The obtained results from this study provided insights that were used to plan an EBSA program 456 tailored to the needs and abilities of women from under-resourced communities.

The numerous facilitators barriers identified in this study are likely to be major contributing factors to the gap between awareness and practice.

The authors provided detailed procedure of their experiments in Methods, the laboratory experiments were well designed and executed. The overall experimental design is sound and solid, the results are convincing in general. 

I encourage the authors change Table 2 into graph to get a better illustration effect.

Author Response

Please see the attachment. Thank you,

Georgina.

Reviewer 2 Report

This article presents a qualitative community assessment to enable the EBSA program to better meet participant’s needs and explore their willingness to enrol in the EBSA program. The data were interpreted through a thematic analysis conducted using NVivo 12 software. The results showed that women were responsible of cooking and shopping for their households. Also participants described their diets and gave examples of foods that they liked and ate frequently; at the same time, they expressed their understanding of healthy and unhealthy behaviours with examples. The conclusions stressed tthat this study in combination with a study where EBSA participants’ experiences within the program were explored was used to generate recommendations for the EBSA program’s providers to optimize it for future interventions.

The article is well written and the analysis is solid. 

Small attention to details could improve the analysis. 

Also the conclusion section could be slightly expanded-the authors could explain how they could continue the present research. 

Author Response

Please see the attachment. Thank you,

Georgina

Reviewer 3 Report

The work presented to me for review is interesting, it touches on the important issue of women's awareness of eating habits. However, by introducing qualitative research, we have a picture of only a subjective assessment of knowledge represented by a discussion group consisting of 6-11 people. We do not know what individual knowledge is.

Additional remarks:

1. Why did the authors not carry out quantitative research as a supplement - it was enough to add a nutritional variety test to the research

2. Why is there such a small group of women - 60 people - are they representative of the cohort?

3. How many newsgroups were there?

4. Did the women in the newsgroups know each other beforehand?

5. How many meetings of each focus group were there?

6. How, apart from bilingualism, was the moderator of the group selected, had she participated in such projects before, was she prepared to conduct the session?

7. What group of researchers listened to the discussion panel recordings?

8. The results lack information on the size of the family, the number of children and this is very important information, especially in the case of eating habits

9. The introduction is very long, it shows that the EBSA project is developed and in the Conclusions the authors write about its changes.

10. Why were the corrections in the EBSA draft not made after the publication of 2020, but only after the results presented in the current work were developed?

11. What changes have been proposed based on this research?

12. The introduction seems to be written at the request of the organization and there is no description of the situation and state of knowledge about nutrition in this region.

13. No exclusion criteria from the group

Work needs improvement

Author Response

Please see the attachment,

Thank you,

Georgina

Round 2

Reviewer 3 Report

1.      Why did the authors not carry out quantitative research as a supplement - it was enough to add a nutritional variety test to the research

This study was part of my PhD thesis together with the previous study published in 2020.

A doctoral thesis is not an article, I know because as a professor I run doctoral students and write articles. A doctoral dissertation is usually at least 70 pages long, while an article is 15-20 pages long, so they cannot be compared. In the dissertation, describing the second part of the research, we simply add another chapter, and in the second article, the whole work is created anew, especially if the time interval is 2 years.

Please, see our previous study: Pujol-Busquets, G.; Smith, J.; Larmuth, K.; Fàbregues, S.; Bach-Faig, A. Exploring the perceptions of women from under-resourced south african communities about participating in a low-carbohydrate high-fat nutrition and health education program: A qualitative focus group study. Nutrients 2020, 12(4).

My PhD thesis was a sequence of studies that informed each other. First, we conducted the study published in 2020, then the current submission (Community assessment for a low-carbohydrate nutrition education program in South Africa: A qualitative focus group study) and two more studies after that. Each study was used to inform the following one.

According to Fetters (2020), mixed methods studies allow the publication of articles reporting quantitative results, qualitative results, and the results of the quantitative and qualitative integration. Considering the space limitations of academic journals, the publication of multiple manuscripts is a strategy that enables the results of each component to be reported in detail. We have followed this strategy to report the findings from the PhD thesis.

Fetters, M. (2020). The mixed methods workbook. Thousand Oaks: Sage.

Accordingly, the first study published in 2020 was a qualitative exploration of the previous Eat Better South Africa program’s participants through focus group discussions. The current submission (Community assessment for a low-carbohydrate nutrition education program in South Africa: A qualitative focus group study) is wholly different from the first publication of my thesis. 

The results of the current qualitative study, together with that published in 2020, helped the Eat Better South Africa program optimize its design to better meet women’s needs and explore potential participants' willingness to participate in the program. We thought that it was essential for these findings to be published alone to highlight potential barriers to implementing low-carbohydrate nutrition education community programs in marginalised low-income settings. This community assessment was done to select a community to implement a pilot intervention, which was evaluated afterwards through a mixed methods approach. Therefore, we believe it is necessary for the results to be published so there is a better understanding of the specific context-related characteristics of the setting where the Eat Better South Africa pilot intervention took place.

The Ocean View pilot study's mixed methodology involved health assessments, questionnaires, focus group discussions, and in-depth interviews. The results consisted of data from the participants' diet assessment before and after the EBSA program and the 6-month follow-up. Furthermore, it explored the changes in health outcomes through blood tests, blood pressure, body measurements and accelerometers. We will submit the mixed methods article for publication once this current qualitative community assessment study is published.

Therefore, this community assessment was done to select a community to implement a pilot intervention which was evaluated afterwards through a mixed-method approach. We included some information in the discussion but as suggested, we have also included one sentence in the conclusions section.

Please see below on line 457: “These findings helped optimized the EBSA program which was evaluated later through a mixed method pilot study.”

I evaluate the work provided to me by the publisher, and in the title, there is no information that this work is a continuation of the previous one. Writing a dissertation is different from writing an article. Article - each one should be an independent part unless I write them as a series and number them as part I and part II, then the reader can find some common information in the previous part. In this case, there is no such information and the article should be an independent whole.

Each scientific work should be an independent whole.

2.      Why is there such a small group of women - 60 people - are they representative of the cohort?

The communities chosen for this community assessment study were Ocean View and Cloetesvile. Both under-resourced communities studied in this project were established as aresult of the Group Areas Act of the Apartheid Government for mixed-race populations who were forcibly removed from their homes in surrounding areas that were given to whites. These communities face high rates of violence theft, and crime, and a high percentage of the population experience direct or indirect consequences of alcohol and drug abuse. Nevertheless, these communities are much more than what the media often portrayed them. It is vital to consider how this portrayal creates stigmatising effects for community residents and reinforces the idea that these communities are inadequate social environments. It is also necessary to recognise that the community struggles have been linked previously with the traumatic experiences of forced removals.

Women from these communities were a representation of the population which EBSA have had been working before. Purposeful sampling is widely used in qualitative research for the identification and selection of information-rich cases related to the phenomenon of interest. Purposeful sampling is a technique widely used in qualitative research for the identification and selection of information-rich cases for the most effective use of limited resources. This involves identifying and selecting individuals or groups of individuals that are especially knowledgeable about or experienced with a phenomenon of interest.

Please see below the references in the article which explain that the sample size is consistent with recommendations about sample size in the literature on qualitative research:

42. Guest, G.; Namey, E.; McKenna, K. How Many Focus Groups Are Enough? Building an Evidence Base for Nonprobability Sample Sizes. Field Methods 2017, 29(1), 3–22.

Please, see other articles which contribute to the sampling strategy:

Guest, G., Bunce, A., & Johnson, L. (2006). How Many Interviews Are Enough? An Experiment with Data Saturation and Variability. Field Methods, 18(1), 59-82. https://doi.org/10.1177/1525822X05279903 / Onwuegbuzie, A. J., & Leech, N. L. (2007). Sampling designs in qualitative research: Making the sampling process more public. The Qualitative Report, 12(2), 238-254. https://doi.org/https://doi.org/10.46743/2160-3715/2007.1636

As mentioned in the article in the inclusion criteria on line 144 onwards:

“Participants were recruited at a community meeting with the help of a non-profit organization working with members of the community. Purposive criterion sampling [41] was used to select participants fulfilling the following eligibility criteria: women; 18 years or older; capable of providing informed consent; able to understand and speak English or Afrikaans; able to attend the focus group meeting. Women who were interested in taking part in the study were given a participant information sheet those who met the eligibility criteria were recruited as part of the study sample.”

However, we did acknowledge the potential bias in line 402 in the current article, please see below:

“A limitation of the study [26] was that neither diet nor socioeconomic status was evaluated quantitatively in these participants. Therefore, it is not clear to what degree this sample represented the financial status of all EBSA participants and their wider communities. However, although they all lived in low-income communities, they may have represented a more affluent part of the community. The results on barriers to following dietary recommendations reveal personal and environmental factors that make it challenging for participants to make dietary changes or to adhere to EBSA dietary recommendations. Women strongly emphasized financial problems, food insecurity, and the cost of healthy foods. It was an expected result, as most of the participants in this study are unemployed. The findings on financial constraints and food cost are in line with those of previous studies [50,51].”

Information about the reason for receiving the group should be in the article, not in the answer to the review. This document will be known only to me and the editors, and readers should be given an explanation. Since the article is addressed to a worldwide audience, not everyone knows about the eviction policy of the country you describe

3.      How many newsgroups were there?

We are not sure we understand this question. There was a total amount of 6 focus group discussion. A study by Guest (reference 42 in the article) analyses revealed that more than 80% of all themes were discoverable within two to three focus groups, and 90% were discoverable within three to six focus groups. Three focus groups were also enough to identify all the most prevalent themes within the data set. These empirically based findings suggest focus group sample sizes that differ from many of the “rule of thumb” recommendations in the existing literature. Please see the references below:

42. Guest, G.; Namey, E.; McKenna, K. How Many Focus Groups Are Enough? Building an Evidence Base for Nonprobability Sample Sizes. Field Methods 2017, 29(1), 3–22.

43. Adler, P. How many qualitative interviews is enough? National Centre for Research Methods Review Paper. 2012. Available online: https://eprints.ncrm.ac.uk/id/eprint/2273/4/how_many_interviews.pdf (accessed 20 December 2020).

The question is simple. Since 60 women participated in the study, how many groups did they create? Was it one group with 60 people or 6 groups with 10 people? How many women were leading these groups? How big were the focus groups?

Again, in response the authors refer to an earlier publication completely groundlessly that publication is already closed and published this is a new document and requires full information

4.      Did the women in the newsgroups know each other beforehand?

As mentioned in line 144:

“Participants were recruited at a community meeting with the help of a non-profit organization working with members of the community. Purposive criterion sampling [41] was used to select participants fulfilling the following eligibility criteria: women; 18 years or older; capable of providing informed consent; able to understand and speak English or Afrikaans; able to attend the focus group meeting. Women who were interested in taking part in the study were given a participant information sheet those who met the eligibility criteria were recruited as part of the study sample.”

For this study, we did not know if women knew each other for the recruitment of the group discussions. We did not consider adding this in the inclusion criteria since the aim of the study was to assess the community members’ willingness to take part in a community educational project. The EBSA sessions were designed to be provided in groups, therefore, gathering information from groups of women was important. To recruit the participants, a community meeting was organized with the help of local organizations.

In the case of qualitative research, especially within discussion groups, the ability to speak in a group is important. If the women knew each other, it was easier for them to start a conversation, especially since the time of the meeting was short. The fact that I volunteer and want to participate in meetings does not mean that I will start a conversation at such a first meeting. Therefore, it is important whether the women knew each other before. If the authors believe that it is irrelevant, please prove it in the paper. This is qualitative research and the assessment is based on the course of the conversation, so the ability to engage in conversation is important.

5.      How many meetings of each focus group were there?

Each focus group discussion was held separately from the others. There were two community meetings, one for each community and six focus group discussions.

This information should be included in the thesis, not just in the answer

6.      How, apart from bilingualism, was the moderator of the group selected, had she participated in such projects before, was she prepared to conduct the session?

As mentioned in the Acknowledgments section in line 480:

“We acknowledge women who volunteer to participate in this study, Dr. Zelda Holtman for moderating the group discussions.”

Zelda Holtman (PhD), was hired to moderate the interviews because of her expertise as a group facilitator and a researcher herself. She was a senior Professor at the University of Cape Town.

Such information should be included in the text and not in the acknowledgments, as it is important to understand the way the group narrates. Dr. Zelda Holtman was not a randomly selected person but a member of the research group

7.      What group of researchers listened to the discussion panel recordings?

As discussed in the Data analysis section in line 171:

FGDs were audio-recorded in the participants’ language (Afrikaans) and were transcribed and translated into English by Cyber Transcription (South Africa). Using NVivo 12, Braun and Clarke’s approach to thematic analysis [44–46] was implemented in six phases [44]. In Phase 1, GPB read the transcripts several times to familiarize herself with the data and identify initial patterns. Based on these readings, in Phase 2, GPB developed a first draft of the codebook, which was subsequently revised by ABF and SF. Phase 3 involved GPB searching for meaning in the data by dividing each code into several sub-codes. In Phase 4, sub-codes were reviewed by ABF and SF and subsequently displayed by GPB as thematic maps. Phase 5 consisted of naming the themes and defining the main concepts of each theme with the total of four. Phase 6, mainly carried out by GPB, KL, and JS, consisted of producing a report that was later shared with the EBSA members and that included recommendations on how to optimize the program for future interventions. Although only GPB was involved in the coding, all the members of the research team reviewed the coded data and agreed on the final categorization and themes to minimize personal bias.”

A professional transcription and translation community called the Cyber Transcription was responsible of listening to the audio recordings. However, KL and JS, listened to the audio recordings to ensure the transcriptions were correct.

What does the abbreviation GPB mean the first time you use it, write the full name and the abbreviation in brackets so that it is clear to everyone what it means. The authors write "in Phase 1, GPB read" - is this a research group? if so, who was it composed of, if it is a computer system, what kind?

8.      The results lack information on the size of the family, the number of children and this is very important information, especially in the case of eating habits

As seen on Table 2 in line 196, some personal and health characteristics from participants were collected. In this study, we did not consider necessary to ask women about their family size, number of children. The EBSA program is a nutrition education program for women to improve their health. The purpose of the program was not to improve children’s dietary behaviours. Therefore, it was not essential to have this information for our study. Our interest was related to women’s health specifically their metabolic health and the diagnosed diseases.

The question about the number of children is not a question about their nutrition, but if a woman has 1 child, she has more time to take care of her own nutrition, the more children, the less time, less financial resources, especially in poor groups. With a larger family and small financial resources, the mother primarily cares about the food of her children and not her own, therefore it is important to assess the nutrition of a woman with information about her family. It is the mother (woman) who shapes the eating habits in the family

9.      The introduction is very long, it shows that the EBSA project is developed and in the Conclusions the authors write about its changes.[1] 

Thank you for this comment.

The information in the introduction was structured to give the readers an extensive and detailed overview of the current South African health situation and the important of social and behavioural determinants in the development of diet-related diseases. Moreover, we found it essential to provide information about the EBSA program and low carbohydrate high fat (LCHF) diets as this study explored participants opinions about both aspects.

Although we understand the reviewers’ comments, the Nutrients journal does not advise a limit of words for its publications. We consider that the information provided in the introduction is necessary and relevant. At the end of the Discussion, we explained the further studies. In the conclusions, we conclude the main highlights of the study, and we express the importance of community assessments to better meet future participants’ needs. As explained before in our answers, we have added extra information about the mixed method pilot study.

10. Why were the corrections in the EBSA draft not made after the publication of 2020, but only after the results presented in the current work were developed?

As mentioned before in the answer of the first question, to optimize the EBSA program to better meet women’s needs, we collected information from previous EBSA participants and from potential EBSA participants. This community assessment was essential to understand the needs of women who had never participated in an EBSA program. Having the two different types of findings helped us redesigned the program considering the challenges and facilitators that previous EBSA participants went through during their experiences in the program. Moreover, it helped us understand potential difficulties and facilitators that women could encounter in their settings when implementing the EBSA program. Qualitative methods are used both to evaluate community interventions and to gather essential information to assess the community needs. We considered that we had both findings before redesigning the EBSA program for the mixed method pilot study.

As mentioned in line 63 in the study:

“The prevention and mitigation of NCDs in these communities may be positively supported through engagement in community-based nutritional education programs [10]. Essential elements for a successful community-based program include a good understanding of the community functions, close collaboration with various community organizations, and the full participation of the people themselves [22]. With these elements in place, nutritional education program providers can map out a course for health improvement by creating strategies to make positive and sustainable changes in communities [9]. Qualitative methods have increasingly been used to assess a target population needs, preferences and to evaluate given education programs [23–25]. Focus group discussions (FGDs) have been used successfully to develop, or adapt, self-management interventions for NCDs especially in low-income populations and minority groups [25]. The informal style and interactive nature of an FGD are particularly appropriate for identifying barriers to care, exploring health beliefs, identifying education needs, and gathering information to improve intervention programs [23–25].”

The authors' answers are contradictory

On the one hand, the authors indicate that the description of the group, methods, etc. were presented in the article from 2020, and now they point to the multiplicity of EBSA tests and the creation of mixed studies.

The work seems incomplete, it is full of ambiguities and partial information, and the article should present the study and the result of these studies with their summary, even if it is a part of a large project, the fragments described must have a logical basis and a justified summary.

The work in its current form is still chaotic and lacks a logical structure. The authors often force the reader to search for relevant information regarding the study group and methodology or results in other articles. The fact that the authors published half of their doctoral thesis in 2020 does not mean that all readers know about it.

11. What changes have been proposed based on this research?

Thank you for this comment. As mentioned in line 441 of this study. Please, see we have added a Supplement File S2 “Recommendations for EBSA to optimize the nutrition education program” below:

“This study highlighted some aspects of the program that participants felt were essential to its success, such as the cohesive support group and some challenges that need to be addressed. Together this study and the study on previous EBSA participants’ [26] provides an important setting to base recommendations on for the EBSA providers. The small and convenient sample used in this study may not represent the entire population base where EBSA wishes to run programmes, thus, limits the generalisability of the results. The results of this study and the previous one [26] were interpreted, and a report was written with some suggestions for the EBSA team. The recommendations were divided into four main topics, namely recruitment, program, follow-ups, and publicity in the community (File S2). These suggestions were aimed at the re-design of the EBSA intervention to be offered in a new community or a past community, that is, Ocean View. The EBSA program was eventually redesigned for a pilot study in Ocean View and evaluated through a mixed-methods design.”

Thank you for the reviewers’ suggestions. We have added all the changes as a Supplement (File S2) for this study, please see below:

In supplementary materials, the entire program can be shown, while in the text attention should be paid to the aspects that have been changed as a result of these studies

Reduce the Introduction, and going deep into EBSA project. Better connect conclusions more centered in the main findings from the qualitative study in the framework of EBSA and how could be applied in the projecte the identified barriers and facilitators

Author Response

Please see the attachment - new responses in green.

Thank you
